# Being "resilient" and achieving "resilience": From governmental discourses to the national research agenda in the contexts of food insecurity and cost of living

Sara Vilar-Lluch[1], Donna Clutterbuck[2], Michael Kranert[3], Dianna Smith[4], Sarah Nield[5], Nisreen A. Alwan[2,6]*

1 School of English, Communication and Philosophy, Cardiff University, Cardiff, United Kingdom, 2 School of Primary Care, Population Sciences and Medical Education, Faculty of Medicine, University of Southampton, Southampton, United Kingdom, 3 Department of Languages, Cultures and Linguistics, University of Southampton, Southampton, United Kingdom, 4 Geography & Environmental Science, University of Southampton, Southampton, United Kingdom, 5 Law School, University of Southampton, Southampton, United Kingdom, 6 University Hospital Southampton NHS Foundation Trust, Southampton, United Kingdom

* N.A.Alwan@soton.ac.uk, nisreen_alwan@outlook.com

**Data Availability Statement:** The data of the GovRC and FundRC datasets is available from the

## Abstract

The concept of 'resilience' is pervasive, permeating academic disciplines and political discourses. This paper considers (i) the construal of 'resilience' in the contexts of food insecurity and cost-of-living in governmental discourses in the United Kingdom (UK); (ii) to what extent the political representations are reflected in research funding calls of UK national funding bodies, thus showing possibility of shaping research agendas; and (iii) to what extent official uses of 'resilience' reflect lay understandings. We are combining a corpus-based discourse analysis of UK governmental discourses and research funding calls with a study of focus group discussions. Representations of 'resilience' are further compared with those available in general English corpora. We are observing a shift in the use of 'resilience': from an individual psychological attribute to a primarily socioeconomic and environmental characteristic. Funding calls construe resilience in relation to communities, reflecting economy and environment adaptability, whereas governmental discourses frame references to individuals in terms of 'vulnerability'. Focus groups reveal divergent conceptions of 'resilience', which may lead to potential misunderstandings. While this variety of uses may be productive in political discourses for rhetorical purposes, there is a need for specificity in shaping research and in public-facing communications.

## 1. Introduction

'Resilience' has become a pervasive term in policy, politics and workplace discourse, so far so that it is described to be 'overused to the point of banality' [1]. The policy and research field of public health and household food insecurity is no exception to this, as research calls wanting to tackle these social issues seem to use 'resilience' as an umbrella term for any positive

following repository: https://eprints.soton.ac.uk/ using DOI https://doi.org/10.5258/SOTON/D3322. The focus group data is available on request provided ethics committee approval for sharing this anonymised data is granted. To request access conditional on approval, please email rgoinfo@soton.ac.uk.

**Funding:** This work was supported by a University of Southampton seed funding for interdisciplinary research.

**Competing interests:** The authors have declared that no competing interests exist.

outcome. Media, politics and academic research trends to focus on 'resilience' have already been highlighted by McGreavy [2]. 'Resilience' can be regarded as a keyword in public discourse, a socially and culturally significant term that reflects discourses at a certain time [3]. Blommaert [4] argues that discourses have complex textual trajectories, in which texts (and images) are 'repeatedly decontextualised and recontextualised' acquiring new meanings in the process. This study shows how this is particularly true for 'resilience', with origins in the scientific domain and later recontextualised into governmental discourse, thus turning from a category of scientific analysis into a category of everyday experience [5].

In the context of the United Kingdom (UK), the pervasiveness of the concept has extended into the publicly funded research agenda. However, as the term is often not specifically defined by social actors, be that politicians or funding bodies, it becomes vague and heavily dependent on the orders of indexicality, systems of 'social meanings to which people orient when communicating' [4].

For groups of different social actors, the term indexes different discourse histories, variable even by academic discipline. For example, in the context of health, 'resilience' is often used to signify individuals coping with difficulty and crisis [6], while in physical and natural sciences it often signifies robustness and adaptation in people's environments [7]. The question is if 'resilience', without specific definitions being provided, hinders communication between policy makers, the public, researchers, and within interdisciplinary research teams. There also seems to be an apparent correspondence between the increasing uses of 'resilience' in research policy and its recent abundant use in political agenda setting.

We aim to shed a light on this dynamic by examining the meaning of 'resilience' in the contexts of household food insecurity and cost-of-living crisis in the UK by analysing its use in governmental discourse, research funding calls and the social community in focus groups. The recovery from the global Covid-19 pandemic dramatically intensified the cost of living in the UK [8]. Economic pressures from the global recovery of the Covid-19 pandemic and the disruption of supply chains caused by the Russian-Ukrainian war have been reported globally. In the UK context, the aftermath has been particularly severe. In March 2023, the rate of increase in food prices (19.1%) was reported to be the highest since the economic stagnation of 1977, and by May 2024 it had risen by 30.6% since May 2021 [9]. The severity of food insecurity among deprived communities has risen the alarm of widening health inequalities across the country [10].

In this paper, we operationalise ideas from Critical Policy Discourse Analysis [11] with methods from corpus assisted discourse analysis to examine institutional and lay understandings of 'resilience' within the British public by addressing these research questions:

1. What are the academic and political origins of the concept 'resilience'?

2. Do general British English corpora (large collections of digitised texts representative of a language) show any changes in the use of 'resilience' over time?

3. Can we observe any specific patterns of use of the term 'resilience' for governmental communications and funder calls as compared to general use, particularly in relation to the contexts of cost-of-living crisis and food insecurity?

4. Do members of the public associate 'resilience' with the contexts of cost-of-living crisis and food insecurity? What meanings do they orient to otherwise?

In what follows, we first discuss our methodology and data. We then provide a historical contextualisation of resilience discourses, present the results of the corpus analysis and the focus groups, and draw some conclusions for the policy field.

Following substantial changes to government policy, such as the Health and Social Care Act 2012(www.legislation.gov.uk/ukpga/2012/7/contents/enacted) and the Localism Act 2011 (www.legislation.gov.uk/ukpga/2011/20/contents/enacted), there is a notable shift away from central government intervention to decision-making led by locally appointed groups and governments. While this is framed as providing greater choice to localities in developing their priorities and policies, it can also be seen as devolving responsibility for addressing inequalities. This concurs with the use of 'resilience' as the desired outcome for communities, inferring less (central) intervention is needed.

## 2. Methodology

### 2.1 Critical Policy Discourse Analysis

Critical Policy Discourse Analysis (CPDA) underpins policy analysis with linguistic methods from Critical Discourse Analysis (CDA) [11]. Policy analysis has a long tradition in social sciences; originally mainly positivist and technocratic, current studies focus on language and communicative practices [12]. CDA has traditionally considered social problems such as discriminatory practices and how normative ideologies are enacted in discourse (e.g., [13]). CPDA aims to provide critical policy analysis with systematic linguistic tools, sometimes referred to as 'text-oriented discourse analysis'. In order to examine patterns of use and meaning attributions to the term 'resilience' in governmental and funding calls discourses, this paper adopts a corpus-based approach to discourse [14]. This analysis is further complemented with a qualitative examination of exploratory focus groups discussions to account for public understandings.

### 2.2 Corpus-based discourse analysis

Corpus-based discourse analysis involves the adoption of corpus approaches to the study of large collections of texts by combining quantitative examinations with qualitative text analysis —e.g. [15, 16]. Corpus-based approaches to the study of discourse [14], or corpus-assisted discourse analysis (CADS) [17], have made it possible to apply linguistics to other areas of the social sciences concerned with communication. We adopt a corpus-based approach to examine the use of 'resilience' in governmental discourses and UK national research funding calls and identify any shared patterns of meaning between these two domains (research question 3). Two specialised corpora were built for the purpose of this study: the Government Resilience Corpus (GovRC) and the Funders Resilience Corpus (FundRC) (Table 1).

The GovRC includes documents retrieved from Gov.uk applying the summons *"food insecurity" AND "resilience"*, and *"cost of living" AND "resilience"*, filtering by "news and communications" content type and date of publication, including texts published from 1 January 2020 to the date of data collection (22 June 2023). The summon *"food insecurity" AND "resilience"* retrieved 50 documents, and *"cost of living" AND "resilience"* retrieved 79 documents, from which 8 documents were discarded after inspection (one document had already been retrieved with the "food insecurity" search, five documents constituted tribunal decisions and were not relevant for this study, and two documents only included "resilience" as part of the "notes to editors" instead of the original communication).

**Table 1. Corpus overview.**

| Corpus | Documents | Tokens | Words |
|---|---|---|---|
| Government Resilience Corpus (GovRC) | 121 | 211,098 | 185,614 |
| Funders Resilience corpus (FundRC) | 78 | 214,884 | 191,293 |

The FundRC was built searching in the research funding database Research Connect all the UK Research and Innovation (UKRI) funding calls dating from 1 January 2020 to 22 June 2023 that included the term "resilience". Since Research Connect only offers summaries of the calls, usually with changes of expression, the texts were retrieved from the original websites of the calls (provided at the database). The FundRC comprises 78 documents, including descriptions of funding applications (37 documents), and descriptions of 'areas of interest' and research programmes (41 documents). Specialised corpora aim at being representative of a particular register or linguistic variety, for this study the UK governmental and academic communications around resilience. Therefore, uses of 'resilience' observed in the specialised corpora (GovRC, FundRC) cannot be extrapolated to the general British English. Uses of 'resilience' in general British English (research question 2) were examined in two reference corpora: the British National Corpus (BNC), and the English Web enTenTen20 (enTenTen20), both of them available in Sketch Engine. The BNC comprises ~100 million words of written and spoken language from the 20th century (1991–1994) (for further information about the BNC, see www.natcorp.ox.ac.uk/corpus/index.xml?ID=numbers). Since the latest version of the BNC (BNC2014) is not available, to date, in Sketch Engine [18], the software employed for the corpus analysis in this study, the analysis of 'resilience' in the BNC was complemented considering the enTenTen20, which comprises 36,561,273,153 words of texts collected from website domains whose states have English as official language until January 2021 (for further information about the English Web enTenTen, see www.sketchengine.eu/ententen-english-corpus/). Since our specialised corpora only comprise British English, only the UK domain (.uk) sub-corpus of the enTenTen20 was considered, which comprises 3,420,336,927 words and constitutes 7.93% of the whole corpus. Comparing the uses of the terms examined in this study ('resilience', 'resilient') in our corpora (GovRC, FundRC) with those of the reference corpora (enTenTen20 and BNC) makes it possible to examine whether the characterisations observed in the governmental and funders texts are unique of the registers considered, or whether they reflect a more general trend of use within the British English language.

The analysis adopted corpus tools commonly used in corpus-based approaches: keywords, collocations and concordances. Keywords are words that occur "with unusual frequency" in the corpus under study "by comparison with a reference corpus of some kind" [19], thus making it possible to identify the most salient themes of our corpus. Sketch Engine adopts the metrics of 'simple maths' to calculate 'keyness' score and retrieve keywords. Simple maths involves the comparison of the normalised (relative) frequencies of the words in the focus corpus with those of the reference corpus and makes it possible to compare between corpora of different sizes. Specifically, the simple maths formula to calculate keyness score reads: $(fpm_{focus} + N) / (fpm_{ref} + N)$—"$fpm_{focus}$" and "$fpm_{ref}$" are the normalised frequencies (per million) of the focus and reference corpora respectively, and "N" the smoothing parameter. By default, $N = 1$; $N > 1$ (e.g. 10, 100. . .) allows us to focus on higher-frequency words, and $N < 1$ (e.g., 0.1, 0.01), allows us to focus on lower-frequency words. For this analysis, $N = 1$ (for a discussion on keyness metrics, see [20]). Keywords may be single-token items or multiword expressions ('terms'). For this analysis, we considered the first 100 terms (ranked by keyness score) for both the GovRC and FundRC, and classified them into themes defined by checking their use in context with the concordance tool. The concordance tool retrieves strings of text displaying the expression analysed in context (i.e. it shows the co-text or textual environment of the word or a phrase), making it possible to examine how it is used, to identify linguistic patterns, and to infer evaluations [17]. We acknowledge that the 100 terms cut-off is arbitrary, but it allowed us to identify the most salient topics for each dataset and reach saturation–i.e. the point in which new themes do not emerge. Appendix 1 in S1 File provides an overview of the first 50

keywords ('terms') of each corpus classified by theme. Keywords were retrieved using the.uk domain sub-corpus of the enTenTen20 as a reference corpus.

Collocations are words that co-occur together with a frequency higher than chance [21]; for this study we considered the terms that collocate with 'resilience' and 'resilient'. Collocates were extracted with the WordSketch tool. WordSketch organises the collocates into grammatical categories, thus providing an insight into how the collocates are related to the node term ('resilience', 'resilient') and function in context. In particular, we were interested in those grammatical patterns that allowed us to identify: types of resilience (collocates pre-modifying 'resilience'); entities that may be attributed resilience (noun collocates modified by 'resilience'/ 'resilient', subjects of the phrase 'be resilient', or collocates for the prepositional phrase 'resilience of'); entities that may be perceived as shocks in need of resilience (collocates for the prepositional phrases 'resilient to', 'resilience to'); actions associated with resilience (verbs collocating with 'resilience' as object); and qualities or entities commonly associated with resilience (phrases 'resilience and/or', 'resilient and/or'). Appendix 2 in S2 File provides an overview of the collocates for 'resilience' and 'resilient' classified by grammatical pattern. Sketch Engine measures collocation strength (how typical a co-occurrence is) with logDice [22]. The maximum theoretical value is 14 (absolute co-occurrence of two words), and negative values mean that the co-occurrence of the terms is not statistically significant. LogDice is not affected by corpus size, thus allowing us to compare different corpora.

## 2.3 Focus groups

Focus groups facilitate direct insights from the perspective of the public or the particular community of interest, making it possible to examine how participants talk, and potentially think, about the topics under study [23]. From a discourse-linguistic perspective, focus groups can be understood as metadiscourse, showing how language users reflect on language. Understanding how the social community conceives 'resilience' is essential to identify misunderstandings triggered by conflicting conceptions of the term between official uses and common knowledge.

We carried out two exploratory focus groups that addressed two main research questions: (i) What do people understand when they hear 'resilience' or 'resilient'? And (ii) How do they see these terms applying to their lives?

Two one-hour focus groups were held in December 2023 and were facilitated by DC via MS Teams. The focus groups comprised six participants overall. Participants were recruited by an email invitation advertising the study to people who had previously contributed towards patient and public involvement (PPI) activities for the University of Southampton. Since we were interested in collecting lay perspectives on resilience, none of the participants recruited reported professional or academic expertise on the subject during the recruitment process. Other demographic information was not collected from participants for this study. Interested individuals were encouraged to contact the research team to receive study information, and invited to take part in a focus group to discuss the meanings they associate with 'resilient' and 'resilience'. Informed consent was obtained from each participant prior to the start of the focus groups, and participants were offered a £25 shopping voucher as a reward for their time. The study was approved by the University of Southampton Ethics committee (ref. 89082). Group conversations were directed using the open questions displayed in Table 2. Questions were purposefully designed to be as open as possible in order not to precondition responses and obtain participant's genuine views on the meaning of the concept 'resilience'. Follow-up questions were used to elicit more information when required (e.g. if participants did not understand the question as it had been originally formulated—see 'example prompts', Table 2). Absence of questions with explicit references to the topics of cost-of-living crisis and

**Table 2. Topic guide for focus group discussion.**

| Example discussion questions | Example prompts |
|---|---|
| 1. What does the term resilience mean to you? | How do you interpret the term resilience? Is resilience a useful term to use? |
| 2. How would you describe resilience? | Do you have any examples of resilience? What do you think of when you hear the word resilience? |
| 3. Where do you usually hear the term resilience? | In what context? Who usually uses the term resilience? Is it used in a positive or negative way? / supportive or pressure-inducing way? |
| 4. What does it mean to be resilient? | How do you interpret being resilient? Is being resilient a useful phrase to use? Do you think it has individual or collective meaning? |
| 5. How would you describe 'being resilient'? | Do you have any examples of 'being resilient'? |
| 6. Where do you usually hear the phrase 'being resilient'? | Do you hear it in different places to where you hear the term resilience? Who usually uses the phrase 'being resilient'? Is it used in a positive or negative way? / supportive or pressure-inducing way? |
| 7. Are the terms resilient or resilience helpful or unhelpful? | Why? What would make them more helpful? |

food insecurity allowed us to determine whether participants would associate resilience to these societal crises in the first place, as the UK societal context of the time would have suggested. Specifically, questions around context of use of the term and examples (Q3, Q5 and Q6, Table 2) were designed to act as prompts to elicit any associations between resilience and societal and economic conditions. Discussions were audio recorded and transcribed using the MS Teams transcription facility. Transcriptions were revised in full to amend any inaccuracies from the software and anonymised. Transcripts were coded inductively in NVivo (Version 13, 2020 Release 1.0) adopting Braun and Clarke's [24] approach. These codes were used to form the main themes reported in Section 5.

## 3. A discourse history of 'resilience'

### 3.1 Origins and operationalisation of 'resilience'

The concept of 'resilience' can be traced back to the Latin term 'resilire' ('to rebound'), which appeared in early legal terminology to restore the parties to their original legal position (e.g., following a breach of contract) (see [25] for a developed historical account and [26]). The concept was later adopted in the science field, notably within mechanics and material science to describe the capacity of materials and processes to withstand stress, and was associated with material properties such as stiffness, pliability or elasticity [27, 28].

In the early twentieth century, 'resilience' emerged in medicine to refer to the ability of bodily organs to self-regulate and maintain stability (see Cannon [29], who coined the term *homeostasis*). Later at the 1950s, 'resilience' gained prominence in psychology. Key proponents include Garmezy [30], who focused on schizophrenia and stress resistance, competence and resilience, and Bonanno [31], who identified psychological resilience as central to grief and trauma reactions. Resilience has subsequently developed as a dominant theme, including studies on adolescence [32], family [33] or community [34].

From psychology, 'resilience' pervades the wide spectrum of the social sciences—see, for example, Ledesma [35] on resilient leadership within management, or the new concept of resilient property [36]. This move beyond the scientific field to the socio-economic realm appears to have increased political interest in the opportunities it offers in formulating policy which looks to the responsibility of the individual. A further discipline that has adopted 'resilience' is ecology, particularly in relation to adaptation to climate change and other natural phenomena (e.g. [37]).

A relatively recent approach to the conceptualisation of resilience is the one proposed by Resilience Engineering (e.g. [38, 39]). Following Resilience Engineering, a system (an organisation, group or individual) is resilient if it has the capacity to effectively "adjust its functioning prior to, during, or following changes and disturbances, so that it can continue to perform as required after a disruption or a major mishap, and in the presence of continuous stresses" [38]. Adjustment to unexpected changes to continue functioning in the new conditions is associated with the ability to *respond* to changes, but also to *monitor*, *learn* and *anticipate* future challenges [39]. In contrast to the early uses, which would associate resilience with the recovery of a previous state or the capacity to withstand stress, Resilience Engineering highlights resilience as developing capacities for adjustment to continuous environmental changes.

These historical considerations highlight 'resilience' as a kaleidoscopic concept, which far from bearing a unified meaning shows a diverse array of uses and semantic nuances according to field of application. This divergence, we argue, is what makes the use of 'resilience' a particularly prolific concept in the political arena, but also what may carry difficulties in shaping research and communicating with the public. In this study, we follow the more holistic definition provided by Resilience Engineering, which conceptualises resilience as adaptative property of individuals and social groups that emerges through constant interaction with the environment or societal context.

### 3.2 Resilience in governmental discourses

'Resilience' as political keyword has been linked to US President Obama's reactions to Hurricane Sandy [2, 40]. Obama called citizens to be resilient after the natural disaster, construing 'resilience' as an 'ability to cope, no matter how dire the circumstances' [2]. While references to 'resilience' and being 'resilient' had already risen in popularity under the administrations of Nixon and Bush, it was under Obama's mandate that the concept acquired the current prominence in the US governmental discourses [40]. During Obama's time in office, 'resilience' was associated with valued characteristics, including multiculturalism, creativity or a flourishing civil society *inter alia*, and gained a strong positive deontic meaning. Selchow's [40] study of the use of 'resilience' and 'resilient' in Obama's speeches and the National Security Strategy 2010 further demonstrates that that the term was 'constructed as an abstract entity'. The abstract quality of the concept makes it difficult to dispute its meaning, hence turning it into a powerful symbolic tool to legitimise a broad range of political programmes.

More generally, 'resilience' has been described as entrenched in neoliberal ideology [1]. In neoliberal contexts, 'resilience' often interpellates the individuals, communities and private sector as responsible social actors and asks for active citizenship, as opposed to the state organising the social and economic wellbeing [1]. Following Joseph [1], this understanding of resilience is backed up by US and UK government documents, as well as EU policy documents, which have contributed to the proliferation of its use.

### 3.3 Resilience in the context of household food insecurity and cost-of-living crisis

The challenging situation of household food insecurity has increased in prominence in UK media and policy debates since the 1990s. Household food insecurity indicates the inability of a household to access a sufficient quality diet through socially acceptable means [41]. Recent UK data shows that food insecurity is at 15% of the population [42]. The primary response to food insecurity in households includes state intervention such as Healthy Start vouchers and Free School Meals for the lowest income groups. However, the reality is that often the main safety net is food aid such as food banks (e.g. Trussell Trust). These charitable organisations

offer food parcels to households referred to them through schools, social workers or other charities. The cost-of-living crisis in the UK has seen yet more demand for food aid at a time when donations to food banks are decreasing, however, these were ever only meant to be short-term 'solutions' to a household crisis, understood as inability to afford food. In the face of high and rising basic living costs, food is often the most flexible element of a household budget and so we see food insecurity emerging when households face potentially longer lasting financial crises.

Showing resilience when confronted with an inadequate income is difficult. The process of accessing food aid or other support in the UK, such as Free School Meals or benefits, including Universal Credit, requires engagement with state agencies and acknowledging vulnerability, which can be especially problematic for households with children due to concerns about state intervention via social services [43]. Human geography has explored the unsatisfactory expectation of food aid as a core response to household food insecurity [44] and the notion of resilience, which has been taken up in geography after origins in ecology and sociology [45]. Within geography, there is an awareness of recovery following disruption as neoliberal response, and the necessity of acknowledging the role place or context play in expectations of resilience [45]. Wiles picks up this theme of how health geography specifically may contribute to the discussion around resilience by showing the potential for critical contributions to the literature, noting that resilience cannot be the responsibility solely of individuals, failing to note the constraining circumstance of place/context to describe situated resilience [46].

## 3.4 Resilience and Covid-19

By May 2021, a year into the global Covid-19 pandemic, global food prices had increased by more than a 40% since May 2020, while food prices in local markets were conditioned by a cluster of factors including supply and demand, transportation costs, governmental policies and income levels [47]. The global pandemic led to considerations of how societies can be more resilient to crises (e.g. [48–51]). Particular attention has been paid to the resilience of rural communities, understood to experience higher demographic pressures and limited industrial diversity and policy resources which may exacerbate the impact of the crisis [52], and to the context of food insecurity. Food insecurity is regarded as a marker of wealth distribution in a society, economic redistribution and political accountability [50]. Thus, the sharp increase in food insecurity during the pandemic attested systemic weaknesses in food, social and economic systems, and in many contexts evidenced an unequal distribution of societal risk, with some populations being significantly more vulnerable than others, such as in the UK [51, 53]. In the UK context, the global health crisis is sometimes regarded as part of a broader 'permacrisis' or 'polycrisis', i.e. "a state of ongoing disruption to political, economic and social life" which in the UK includes Brexit, years of austerity policies and the crisis of the National Health System (NHS), which merge with the global health and financial crises, the war in Ukraine and increase in energy prices, and climate change [53]. Against this backdrop, the pandemic has widened health inequalities, has evidenced the weakness of relying on food charities and the voluntary sector as primary response to poverty in the UK context [48, 54], and has revealed general global lack of preparation for responses balancing the trade-off between health and economy [49].

Resilience to crisis, or the ability to absorb the shock, reorganise and recover while undergoing change [50], has been attributed both to the severity of the shock and the measures adopted [49]. Studies on the impact of, and responses to, the Covid-19 pandemic have highlighted the importance of market diversification and promotion of local food systems (e.g. food cooperatives, community supported agriculture) and businesses to empower local

communities, diminish over-reliance on imported food and increase the resilience of the food sector [51, 52, 55], and the need for governmental support [54], including recommendations to strengthen "social safety nets and income floors" such wealth redistribution measures or a Universal Basic Income [50].

# 4. 'Resilience' in governmental and research funding bodies' discourses

## 4.1 Overview of the corpora

**4.1.1 Main themes.** The Government (GovRC) and funders (FundRC) corpora comprise different textual registers (press releases and political speeches, and funding calls respectively) and, as such, involve different fields of activity, communicative goals and audiences [56]. A cursory examination of wordlists of the top-50 lexical terms of each corpus (i.e., excluding grammatical words such as prepositions, conjunctions, determiners, and variable terms such as pronouns) evidences the differences in the field of register (Table 3). The UK features as the most common geographical reference in both corpora. Consistent with the communication goal, the FundRC features references to funding applications (*research*, *application*, *cost*), while the GovRC includes references to support provided and the population. The economy and, in a lesser extent, the environment emerge as the most salient themes of the GovRC (*work*, *energy*, *business*, *climate*) (Table 3).

Examining the keywords provides a more refined overview of the 'aboutness' of each corpus (Appendix 1, Table 7 in S1 File). 'Economy' (e.g. *supply chain*, *inflation target*) and 'climate and environment' (e.g. *net zero*, *climate change*) feature as the most recurrent themes in the governmental discourses (GovRC), together with 'food' (e.g. *food insecurity*, *food security*) and 'cost of living crisis' (e.g. *energy bill*, *cost of living*) themes. The topics of food insecurity and cost-of-living crisis were intentionally included in the query to build the specialised corpora, and references to these themes should thus be expected. The recurrent references to the environment and economy are remarkable, suggesting that references to resilience tend to appear in relation to these matters in governmental discourses. The double-coding of some keywords as providing references to energy and as being used in relation to economy (*energy security*, *energy market*) or cost-of-living crisis (*energy bill*, *energy price*) evidences the energy crisis as one of the main shocks that need resilience, both from the individual or household level, and from the national economy (see Appendix 1, Table 7 in S1 File).

In consonance with the recurrent use of terminology related to funding applications revealed by the wordlists (Table 3), 'research' is the most prominent theme identified for the

**Table 3. Frequent lexical items ranked by raw frequency.**

| GovRC | | | | FundRC | | | |
|---|---|---|---|---|---|---|---|
| Rank | Item | Raw freq. | Normalised freq. per million | Rank | Item | Raw freq. | Normalised freq. per million |
| 21 | uk | 933 | 4419.75 | 13 | research | 2028 | 9437.65 |
| 22 | support | 860 | 4073.94 | 19 | uk | 1177 | 5477.37 |
| 26 | people | 700 | 3315.99 | 21 | application | 1089 | 5067.85 |
| 27 | work | 685 | 3244.94 | 28 | support | 853 | 3969.58 |
| 34 | government | 591 | 2799.65 | 33 | organisation | 718 | 3341.34 |
| 35 | energy | 585 | 2771.22 | 37 | work | 674 | 3136.57 |
| 38 | help | 532 | 2520.16 | 41 | opportunity | 592 | 2754.97 |
| 45 | climate | 481 | 2278.56 | 42 | cost | 592 | 2754.97 |
| 46 | country | 477 | 2259.61 | 44 | innovation | 554 | 2578.13 |
| 47 | business | 459 | 2174.35 | 45 | ukri | 552 | 2568.83 |

**Table 4. Raw and relative frequencies (pm) of "resilience" and "resilient" in the corpora considered.**

|  | enTenTen20 (UK sub-corpus) | | BNC | | GovRC | | FundRC | |
|---|---|---|---|---|---|---|---|---|
|  | hits | RF (pm) | hits | RF (pm) | hits | RF (pm) | hits | RF (pm) |
| "*resilience*" | 50,622 | 14.80 | 225 | 2.34 | 159 | 856.62 | 331 | 1,730.33 |
| "*resilient*" | 22,790 | 6.66 | 212 | 2.205 | 90 | 484.88 | 152 | 794.59 |

keywords of the funders' corpus (see Appendix 1, Table 7 in S1 File)—41 out of the first 50 'terms' of the FundRC refer to research-related matters, such as types of documentation and individual roles involved in applications. Remarkably, if register-specific jargon is excluded, the most salient theme is 'climate and environment' (e.g. *net zero*, *climate change*) and, in a much lesser extent, economy (*supply chain*, *national capability*), which reflects the themes identified in the governmental discourses.

**4.1.2 Uses of 'resilience'.**　Table 4 shows the number of occurrences of 'resilience' and 'resilient' in our corpora and the reference corpora adopted for the study, including both raw and relative (RF) frequencies (per million words). The disproportionate number of occurrences of the terms in the GovRC and the FundRC is not remarkable, since for the purpose of the study the datasets were intentionally built with texts that referred to 'resilience'. The sharp increase in uses of 'resilience' and 'resilient' in the enTenTen20 UK sub-corpus in relation to the BNC (around 7 and 3 times higher respectively as reflected in RF) is of interest. In Section 4.3 we show that collocates for 'resilience' and 'resilient' reflect more similarities in use between the focus corpora (GovRC and FundRC) and the enTenTen20 than with the BNC.

Although both reference corpora aim at being balanced (i.e., with a proportionate distribution of the linguistic and situational variables considered, see [57]), frequency of occurrence and topicality could be partly conditioned by the registers included in each of the reference corpora. For example, the enTenTen20 is annotated by genre (e.g., blog, fiction, legal, news) and topic (e.g., culture and entertainment, travel and tourism, religion) (see www.sketchengine.eu/ententen-english-corpus/). The use of direct and reported speech in journalistic discourse could facilitate the inclusion of wordings from political speeches, leading to better reflect the uses observed in GovRC. The different frequencies can also reflect an increased concern for 'resilience', as evidenced in the use of the term in public discourses, and a change in the use of the term throughout the last 20 years in British English (the BNC available in Sketch Engine dates from 1994, whereas the enTenTen20 dates from January 2021).

Table 5 shows the distribution of 'resilience' and 'resilient' across the BNC and the enTenTen20. 'Relative density' shows the frequency of the terms in the specific genre or topic considered in comparison to the whole corpus; more than 100% indicates typicality (the term presents a higher frequency in the particular genre/topic than in the whole corpus). References to 'resilient' and 'resilience' predominate in written registers, notably news ('written books and periodicals' for the BNC, 'news' for the enTenTen20). In enTenTen20, the most recurrent topics include science, reference, business and health, aligning with the historical development of the concept (Section 3.1). In comparison, the distribution of the terms in our corpora appears relatively even—Juilland's D for 'resilience' and 'resilient' in the GovRC is 0.8881 and 0.8528 respectively, and in the FundRC is 0.8625 and 0.7818 respectively (calculated with the software #LancsBox 6.0, [58]). Juilland's D falls between 0 (very uneven distribution) and 1 (very even). Gries' *DP-norm* (normalised deviations proportions, sometimes preferred in corpus linguistics, see [59]), which falls between 0 (very even distribution) and 1 (very uneven), also provides moderately even distributions, albeit comparatively less—*DP-norm* for 'resilience' and 'resilient' in the GovRC is 0.4914 and 0.5519 respectively, and in the FundRC is 0.3885 and 0.4387 respectively. The moderately even distributions of our corpora can partly be explained by

**Table 5. Distribution of lemma = resilience|resilient in the BNC and enTenTen20 according to genre and topic.** RF, Relative Frequency (normalised pm, per million words).

| Corpus | Genres | Frequency | RF (pm) | Relative density |
|---|---|---|---|---|
| **BNC** | Written books and periodicals | 400 | 4.41 | 113.38% |
| | Written miscellaneous | 31 | 3.70 | 95.12% |
| | Spoken context-governed | 5 | 0.72 | 18.63% |
| | Written-to-be-spoken | 1 | 0.68 | 17.43% |
| **enTenTen20** | *None* | 72,096 | 21.64 | 100.81% |
| | News | 1,065 | 16.98 | 79.10% |
| | Legal | 145 | 10.27 | 47.86% |
| | Blog | 84 | 12.00 | 55.92% |
| | Discussion | 22 | 4.80 | 22.35% |
| | **Topic** | **Frequency** | **RF (pm)** | **Relative density** |
| **enTenTen20** | *None* | 53,298 | 11.64 | 54.22% |
| | science | 5,290 | 29.87 | 139.17% |
| | reference | 5,227 | 11.35 | 52.89% |
| | business | 2,655 | 10.75 | 50.09% |
| | sports | 1,767 | 7.04 | 32.78% |
| | health | 1,542 | 12.24 | 57.02% |
| | arts | 1,254 | 3.33 | 15.51% |
| | technology | 977 | 7.80 | 36.34% |
| | home | 692 | 6.04 | 28.16% |
| | society | 362 | 1.82 | 8.46% |
| | recreation | 182 | 1.08 | 5.02% |
| | news | 167 | 6.31 | 29.41% |
| | games | 35 | 4.46 | 20.77% |

register-specificity (each corpus exclusively contains texts of one register) and by the search terms used for corpus building.

The next sections examine how resilience is characterised in each corpus and consider whether these main themes feed into the portrayals of resilience offered by governmental and funding bodies.

## 4.2 From the individual to the community

The BNC, the oldest of the corpora examined, mainly feature uses of 'resilience' in relation to human psychology and displays a generalised positive prosody. The construal of resilience in relation to the individual subject is especially noticeable among the modifiers of 'resilience', the nouns modified by 'resilience', and conjunctive structures ('resilience and/or') (e.g., *fortitude*, *sweetness*, *hardness*, *indomitable*, *stoicism*, *defiance*, *courage*, *adaptability*, *firmness*, *tenacity*—see Table 8 in Appendix 2 in S2 File and examples 1–2; all examples have been retrieved with the Good Dictionary Examples (GODEX) tool of Sketch Engine, which considers features such as sentence length or use of complex vocabulary to identify sentences that are illustrative and easy to understand). References to the individual character are also salient among the collocates of 'resilient' in the BNC corpus (e.g., *self-belief*, *robust*, *self-reliant*, *tempered*, *tenacious*—see Table 9 in Appendix 2 in S2 File).

[1] These disabled children sometimes <u>showed **resilience** and</u> determination, calling names back or, better still, ignoring unkind comments. (BNC)

[2] Qualities such as **resilience**, adaptability, creativity, tolerance and compassion are the real basics for employment and for general living. (BNC)

The understanding of resilience as an individual characteristic also emerges from the enTenTen20 (UK sub-corpus), albeit in much lesser extent. For example, nouns modified by 'resilience', or terms collocating with 'and/or resilience/resilient' include character qualities such as *self-confidence*, *perseverance*, *tenacity* or *self-esteem*, again all positively evaluated (see Tables 8, 9 in Appendix 2 in S2 File and examples 3–4). Examples 1–4 show that resilience evokes positive moral judgements of the individuals, attributing them psychological strength.

[3] Developing **resilience**, self-esteem and ways of resolving conflict can all support students who may be vulnerable. (enTenTen20)

[4] The programme promotes emotional **resilience**, self-confidence and emotional intelligence. (enTenTen20)

In contrast, references to the individual character do not feature in the governmental and funders' corpora. The main themes of resilience identified in the GovRC and FundRC include references to climate change, supply chains, the agriculture and farming sectors, and the economy, the latter particularly notorious in GovRC (see also Appendix 1, Table 7 in S1 File). These themes converge with those identified in the enTenTen20 (UK sub-corpus), which include references to resilience in the contexts of natural environment, food and economy (e.g., *socioecological*, *flood*, *food-system* as modifiers collocating with 'resilience', Table 8 in Appendix 2 in S2 File). The themes are also observed in the enTenTen20 collocates for the prepositional phrase 'resilience of' and noun collocates modified by 'resilient', which feature references to the natural world (*ecosystem*, *woodland*, *reef*), the economy (*sector*), the health system (*GP*, *NHS*) and *communities*, notably *rural*, *coastal*, *urban* and educational (*resilient learning communities*) (see Tables 8, 9 in Appendix 2 in S2 File). The enTenTen20 also shows uses of 'resilience' in contexts of poverty, food insecurity and vulnerability in descriptions of *resilient livelihoods* (examples 5–6):

[5] Our work has focused on food security, **resilient livelihoods**, health, accountable governance, economic justice, energy and climate change. (enTenTen20)

[6] Food and nutrition security can be made more likely by strengthening women's roles in promoting sustainable and diverse diets, **resilient livelihoods**, local food systems and climate-smart agriculture (. . .) (enTenTen20)

In line with conceptions of ecological resilience [37] and Resilience Engineering [38], these uses connote capacity to adapt to adverse environmental and socioeconomic circumstances. The allusion to adaptability echoes uses of 'resilience' observed in the BNC, but while in the BNC *adaptability* mostly refers to the individual (example 2), here the quality is attributed to communities or social groups (e.g., *women* in example 6).

The next section considers how the Government and funders' corpora resonate with these two main understandings of 'resilience'—a characteristic of individuals or of socioeconomic and environmental systems.

## 4.3 Contexts of resilience in governmental and funding bodies' discourses

Governmental and funding bodies discourses reflect two main operational contexts for resilience: in relation to social actors, which include specific communities and the society at large, and in relation to systems such as the natural ecosystem or economy, which prevail in both datasets. The focus on these two contexts depends primarily on the corpus: governmental

discourses feature the highest number of references to social actors, while resilience in the funders' discourses is largely related to the environment and economy-related activities. These differences are mainly attributed to the textual register, notably the communication goal and audiences addressed. The governmental discourses include political speeches and press releases about policies and programmes, which mainly address Parliamentary members or the general population, and references to those social actors deemed to be resilient or in need of resilience are thus to be expected. Social actors are either presented as those that struggle because of the circumstances described and beneficiaries of the schemes presented, or addressed as part of the audience. In contrast, the funders disseminate funding calls for (primarily) the UK academic community, and can be expected to describe areas of research interest for research projects; references to social actors are minimal, included as potential project beneficiaries or as those individuals involved in the research project.

Social actors characterised as resilient were identified by examining the collocates for the prepositional phrase 'resilience of' (Table 6) and the nouns pre-modified by 'resilient' (Table 9 in Appendix 2 in S2 File). Individualised and collectivised references to the social community were further explored by performing searches for *community*, *individual* (noun), *population*, *people*, *family* and *household* pre-modified by 'resilient' and, following observations from the examination of concordances of the collocates, 'vulnerable', at a distance of up to 10 words (Table 6). While 10 words is a far too wide span for identifying terms in pre-modifying relation, usually in distances up to 4–5 words, increasing the span allowed us to identify descriptions of the vulnerable groups and causes of vulnerability.

**4.3.1 Governmental discourses.** Governmental discourses identify as main resilient actors the economy (*economy*, *business*), the agriculture (*crop*), and the population (*population*, *people*, *community*), including references to workers and the health system (*workforce*, *NHS*) (Tables 8, 9 in Appendix 2 in S2 File), but also to the private domain (*household*, *family*) (Table 6). Explicit descriptions of social actors as 'resilient' are scarce, evidencing an avoidance

**Table 6. References to social actors in the FundRC and the GovRC.** RF, Relative frequencies (normalised per 1000 words).

|  | FundRC | | GovRC | |
|---|---|---|---|---|
|  | **Raw freq.** | **RF** | **Raw freq.** | **RF** |
| **"community"** | 338 | 1.77 | 123 hits | 0.66 |
| *pre-modified by "resilient"* | 16 | 0.08 | 3 | 0.016 |
| *pre-modified by "vulnerable"* | 0 | 0 | 5 | 0.026 |
| **"individual"** | 70 | 0.37 | 34 hits | 0.18 |
| *pre-modified by "resilient"* | 0 | | 0 | 0 |
| *pre-modified by "vulnerable"* | 1 | 0.005 | 1 | 0.005 |
| **"population"** | 47 hits | 0.25 | 45 hits | 0.24 |
| *pre-modified by "resilient"* | 0 | 0 | 1 | 0.01 |
| *pre-modified by "vulnerable"* | 0 | 0 | 4 | 0.02 |
| **"people"** | 152 hits | 0.89 | 698 hits | 3.76 |
| *pre-modified by "resilient"* | 0 | | 0 | 0 |
| *pre-modified by "vulnerable"* | 2 | 0.01 | 24 | 0.13 |
| **"family"** | 7 hits | 0.04 | 130 hits | 0.70 |
| *pre-modified by "resilient"* | 0 | 0 | 0 | 0 |
| *pre-modified by "vulnerable"* | 0 | 0 | 8 | 0.04 |
| **"household"** | 2 hits | 0.01 | 146 hits | 0.79 |
| *pre-modified by "resilient"* | 0 | 0 | 0 | |
| *pre-modified by "vulnerable"* | 0 | 0 | 18 | 0.09 |

of overt attributions of resilience to human beings. Instead, governmental discourses allude to vulnerability in depictions of *people* and *households* (Table 6).

The communities identified in the governmental discourses include international populations that receive support from the UK Government. Climate adversities are acknowledged as a main factor leading to resilience (example 7), and women are identified as a particularly affected group within the 'vulnerable communities' (example 8).

[7] . . .drought assistance and resilience program through the Building **Resilient** Communities in Somalia. (GovRC)

[8] . . .it will undermine opportunities for women in **vulnerable communities.** (GovRC)

Collectivised references to *people* are salient in the governmental discourses (Table 6) and denote both international and national populations. References to *vulnerable people* include explicit and implicit identifications of the Covid-19 pandemic, the living costs and food insecurity crises as sources of vulnerability (examples 9–10). *Families* and *households* are identified among the vulnerable groups in the UK context (examples 11–12). Other vulnerable populations include low-income and elderly populations, and people with disabilities (example 11).

[9] To help with the cost of living, we are going to provide significant, targeted support to millions of the most **vulnerable people** in our society: Those on the lowest incomes, pensioners, and disabled people. (GovRC)

[10] . . .£5 million ($6.7 million) to support cash-based food assistance for approximately 123,000 **vulnerable**, food insecure **people**. (GovRC)

[11] A quarter of all UK **households** will receive £1200 of direct help as part of our £37 billion package to assist the most **vulnerable**, with the first cost of living payments already paid out to over seven million **people.** (GovRC)

[12] . . .Support Fund, which doubles its total amount to £1 billion to support the most **vulnerable families** with their essentials over the coming months. (GovRC)

References to *vulnerable people*, *families* and *households* resonate with depictions of vulnerability as the risk of being negatively affected by a hazard [60] and tend to include identifications of the UK Government as provider of financial support. The Government can either be explicitly mentioned as agent of change, as in example 9, where it is construed as Actor in material clauses (processes that involve a doing or happening [61]), or inferred, as in example 11, where it is evoked through the possessive determiner (*our*). Descriptions of support schemes often include references to monetary figures (examples 10–12), which contribute to emphasise the supportive role of the Government.

Contexts deemed to be in need of resilience include references to *climate change* and natural disasters (*draught*) (see Tables 8, 9 in Appendix 2 in S2 File, collocates for the prepositional phrases 'resilience / resilient to'). References to resilience also alluded to adaptability and sustainability (collocates for 'resilience and/or', Table 8 in Appendix 2 in S2 File). Examining concordances for *adapt\**, which includes all derivative suffixes (e.g., *adaptation*, *adaptability*), allowed us to identify the entities and contexts that need adaptation. The search retrieved 117 occurrences (RF = 0.63—RF calculated per 1000 words), and all except one, which referred to exams support, portrayed adaptation in relation to economy (e.g., *adapt to incentive-based regulation*) and climate change, frequently considered together (e.g. 'adapt to' *environmental change*; *international funding for climate adaptation*; *adapting their business for a climate changed world*). Resilience is thus used as synonym of having adaptative capacity to economic and environmental hazards, by opposition to taking precautionary action to mitigate or prevent them.

**4.3.2 Funding bodies' discourses.** The funding bodies portray resilience in relation to supply chains, the environment, technology, the agriculture and farming sectors (*farming*), and the general population (*society*, *UK*). Social actors are mainly referred to through collectivised references—*community* and, less frequently, *people* and *population* (Table 6). As for the governmental discourses, explicit characterisation of social actors as 'resilient' are scarce. References to *community* mainly refer to UK coastal and local populations, characterised as resilient or in need of resilience (example 13) and to research groups (*interdisciplinary communities*). Similarly, *people* is mainly used to identify UK national populations and those involved in research. The references to social actors as resilient in the context of sustainability projects echo characterisations of resilience as adaptability to adverse events and capacity to recover [38, 60].

[13] . . .**resilient** and sustainable UK coastal **communities**. (FundRC)

Environment and health are identified as contexts that need resilience—e.g., (climate) *change*, *geohazard*, *pests*, *disease* feature as collocates for 'resilience and/or' (Table 8 in Appendix 2 in S2 File). As for the governmental discourse, we examined contexts and entities deemed in need of adaptation by performing a concordance search for *adapt**. The search retrieved 124 occurrences (RF = 0.65, RF calculated per 1000 words) and revealed that the funders refer to adaptation mainly in relation to climate change (e.g., *adapt and build resilience to climate change*; *adaptation and resilience to natural hazards*; *adaptation of biodiversity in response to environmental challenges*), and to the infrastructure of the transport system in relation to finding solutions for reducing carbon emissions (e.g., *implementing sustainable*, *low carbon*, *adaptation solutions for resilient transport infrastructure*).

The funders' emphasis on climate change is consistent with the governmental discourses and, more generally, with the enTenTen20, which features explicit references to environment, climate *change*, and extreme *weather*, including *flooding*, *drought*, and *natural*, *environmental* and *climate-related disasters*, and *humanitarian disasters*. The focus on adaptability as means for resilience aims to advance research that contributes to palliate the negative outcomes of climate change and health hazards, particularly among those communities that have traditionally been reported as being exposed to higher pressures, such as rural communities [52]. Indirectly, focusing on adaptability can also contribute to reproduce the governmental inclination for promoting adaptation to shocks as opposed to preventive or more drastic remedial actions to limit the development of stressors.

## 5. Focus groups

Focus groups discussions allowed us to examine lay people's perceptions about resilience. In contrast to the corpus-based analysis of political discourses and funding calls, these discussions do not show how people use the term in authentic contexts, which makes it possible to hypothesise that participants might use it in ways that could differ from their own metalinguistic reflections.

Four main themes emerged from the focus groups' discussions: (i) resilience as the individual capacity to tolerate or overcome challenges; (ii) resilience as an attribute of socioeconomic and environmental contexts; (iii) resilience as a concept susceptible to misinterpretation, particularly due to (iv) resilience is a concept with generational-dependent meaning.

### 5.1 Resilience as individual attribute

All focus groups participants characterised resilience as the individual ability to tolerate or overcome challenges, shocks or stressors without experiencing significant negative impact.

This understanding of resilience as an overtly positive personality trait is consistent with the meanings observed in the BNC. Individual traits identified as indexes of resilience included *stamina*, *strong*, *flexible*, *adaptable*, *survival*, *maturity*, *patience*, *optimism*, and *self-esteem*. Examples 14–15 highlight the association of resilience with individual agency; resilience is identified with the capacity to 'fight' against the challenging situation (example 14) and the attitude of trying to overcome challenges on one's own before seeking help (example 15). The identification of resilience as the restoration of a previous state (example 14) echoes the early understandings of the concept in natural sciences. This emphasis on restoration or maintenance of stability is in tension with the understanding of resilience as adaptation to changes.

[14] (. . .) resilience to me is about being able to somehow <u>fight back</u> or <u>come back to the position that you were</u>, you were at before (. . .)

[15] (. . .) resilience is instead of immediately turning to other people for help thinking in terms of what can I do first, what can I do to help myself?

Occasionally, resilience was also characterised as the ability to understand when help is needed (examples 16–17). While these examples echo the importance of a support system (personal networks, public services) for individuals to be resilient, the ultimate foundation of resilience is placed on the individual, the one that has to understand when and how to seek for help. The identification of resilience with individual psychological and cognitive skills is explicit in example 16.

[16] (. . .) it's using lots and lots of different types of skill (. . .) and being realistic about then whether you can solve a situation and problem yourself or whether you do need external support and help (. . .)

[17] Resilience should be about knowing when to ask for help, not just about the coping on your own.

In line with the identification of resilience as a skill, participants further characterised it as something that needs constant discipline ('it has to be supported and (. . .) nurtured'), thus placing the individual as the one responsible to develop resilience. Once conceptualised as an individual skill that requires perseverance, failure to show resilience can be regarded as a moral deficit. The negative judgement is explicit in example 18 (underlined). In identifying being resilient as a choice ('. . .still does not change. . .'), resilience is depicted as a matter of free will, overriding lack of capacity as explanatory factor. The individual is attributed responsibility over the negative consequences resulted from the lack of resilience, and the possibility of causes out of the individual's control is omitted. This moralising individualistic discourse has the potential of othering those unable to adopt resilient behaviours.

[18] (. . .) if that person does get told in a nice way, and if that person still does not change (. . .) who is going to suffer? That individual. (. . .) if they don't make the effort themselves, while those people who have been supporting there will come a time when they're going to walk away, right, <u>you can't expect anybody</u> to <u>give you help right from day one till the day the person pops off</u>. (. . .)

Some participants explicitly acknowledged the negative moral judgements that may result from appraisals of lack of resilience (example 19) and argued against the appropriateness of applying evaluations of resilience in certain contexts (examples 20–21). Notably, expectations of resilience as individual capacity should not apply to health-related contexts, where failure to show resilience could be perceived as blaming the individual for not being able to overcome difficulties beyond their control.

[19] (. . .) it's a weakness to to not be resilient and resilient means relying on yourself and just getting on with it. (. . .) so it's a weakness to be other than that (. . .)

[20] (. . .) there was (sic.) some things that you can't always expect people to be resilient against (. . .) if there are multiple adversities in someone's life or multiple traumas (. . .)

[21] (. . .) it may feel like it is their being blamed and that they are on their own like (. . .) just get on with it. (. . .) there was a neurologist who I had appointments with and his manner was very much like that. And it was it was really not positive experience (. . .)

## 5.2 Resilience as societal and environmental attribute

Participants understood 'resilience' in relation to economic and environmental policies and infrastructure (example 22). These descriptions are consistent with the uses identified in the enTenTen20 and the governmental and funders' discourses. Together with climate change, participants alluded to the Covid-19 pandemic as a factor that required resilience, and observed that health and environmental difficulties are most likely to impact young populations (example 23).

[22] (. . .) that cop 28 going on now (. . .) it's being used with its hollow words and not being properly broken down into categorization of what that really means for a specific thing to do with halting climate change.

[23] (. . .) you're right in terms of the impact of COVID and how resilient individuals have been (. . .) there will be other pandemics (. . .) but it's also climate change. And I do feel for young people (. . .)

Participants further identified resilience with the context of education. Within the educational context, 'resilience' was characterised as providing emotional support to young people and teaching them ways to overcome challenges and reaching for assistance, and helping those parents that experience financial hardship (example 24).

[24] (. . .) how can we support these young people to become more resilient (. . .) be able to deal with challenges (. . .) how can we make parents become more resilient that they can— some of the challenges in life whether that's unemployment or whether that's things do with housing (. . .) poverty (. . .)

## 5.3 Resilience as a misleading and generational concept

Participants expressed dissatisfaction with the media use of 'resilience' in relation to socioeconomic and environmental contexts due to the often lack of explanations (example 22), which makes it appear as having become a 'buzzword', the latter particularly noted in relation to education and support provisions for those experiencing financial adversities. Thus, while participants considered that resilience can be a useful term and associated it with a positive prosody (e.g., 'I've never heard it used negatively. . .'), they nonetheless noted that explanations or examples should be provided for the term to be fully meaningful and prevent misinterpretations (example 25). Example 26 further alludes the need to provide detailed guidance (*strategies*) so individuals can effectively adopt resilient behaviours.

[25] (. . .) strategies need to be attached and an explanation needs to be attached (. . .) maybe an example depending on when it where it was being used.

Some participants explicitly oppose the blame and (negative) moral judgements that understanding resilience as an individual attribute can evoke in certain contexts. Example 26 suggests that these readings are misunderstandings of the concept, which is understood positively-valanced exclusively:

[26] I really like the phrase being resilient, but (. . .) I think there needs to be explanations attached to it because some people might not totally understand what being resilient is (. . .) If you're saying to them being resilient, that looks like (. . .) get on with it, and that's not what it means at all.

Participants also noted that different age groups interpret 'resilience' differently, observing that while young people may not antagonise resilience with help-seeking behaviours, older generations primarily understand it in relation to material difficulties, the individual capacity of adaptability (example 27) and being independent (example 28).

[27] So how they used what they've got to meet to make what they needed. So it's a different kind of resilience.

[28] (. . .) Whereas son's age, I think it's more about asking for the right help and that's part of your path of being resilient. I think a lot of people in their 80s (. . .) their idea of being resilient is actually the opposite. (. . .) they tried to do everything as much as they possibly can. Resilience and independence is (sic.) interlinked (. . .) with an older generation (. . .) with the younger generation, resilience and support is more interlinked.

While participants associated the different interpretations to age, the ultimate causes alluded are socioeconomic, making it possible to infer that, should material conditions change, younger generations would potentially conceive resilience in similar terms. Example 28 further echoes the tension reflected in examples 15–17, where resilience is associated with both the ability to solve difficulties independently and the capacity to ask for support.

## 6. Concluding discussion

This study has examined the discursive trajectories of 'resilience' by exploring its historical origins, and examining changes in use as reflected in corpora representative of the English language (BNC, and English Web enTenTen20). The study further considered how 'resilience' is adopted in current governmental and research funders' discourses, and how members of the public understand it. Comparing the use of 'resilience' across these three datasets has made it possible to identify correspondences between political discourses and national research priorities, and contrast official uses of 'resilience' in public discourses with accounts of the social community.

### 6.1 Findings

'Resilience' is a public discourse keyword [3] loaded with connotations acquired throughout its gradual adoption by different academic disciplines. From the original legal meaning of 'restoration' or 'to rebound', the meaning of 'resilience' was expanded in material sciences to describe the propriety of materials to resist stress. The medical sciences extended 'resilience' to the realm of the body to designate the capacity of organisms to maintain their stability under changing circumstances. The medical shift brought resilience to the human domain, albeit from the perspective of a biological system. Psychology applied 'resilience' to the human psyche; resilience was identified with the capacity to overcome trauma or grief, and was associated with recovery and the ability to adapt. Eventually, the social sciences and ecology broadened 'resilience' to the capacity of social and natural systems to adjust to changes, expanding the

connotation of physical flexibility to system adaptation to change, in contraposition to stability or permanence.

The historical discursive trajectories reverberate with the general and specialised corpora considered in this study. The corpora show an overwhelming positive valence and reveal different meaning attributions. The comparative study of the collocates of 'resilience' and 'resilient' in the BNC and the enTenTen20 reflects the move from the individual domain in the BNC, where resilience was mainly used in relation to psychological characteristics, to social groups and natural systems in the enTenTen20, with a focus on the economy and environment. The double reference of resilience as both an individual and system characteristic permeates the present social understandings: while governmental and research funders' discourses mainly adopt resilience as an attribute of the socioeconomic and financial systems or in relation to specific social groups, the focus groups showed an orientation towards resilience as a primary individual attribute.

Uses of resilience in relation to the human domain, either as individual faculty or as a characteristic of social communities or human-related activities (e.g. businesses), reflect an association of resilience with agency, highlighting the adaptability, strength and resourcefulness of individuals and communities to overcome challenges. While resilience is a necessary ability of individuals, social groups and organisations to adjust to changes and be able to effectively function in the new state, focus group discussions also showed that the human agency associated with resilience can resonate with moral discourses, which make it possible to associate lack of resilience with inadequate perseverance and, ultimately, reframe it as a moral deficit. Identifications of resilience with individuals are avoided in both governmental and funders' discourses. Instead, the environment and economy themes predominate as salient topics, especially in the governmental discourses. References to social actors in the context of resilience are more frequent in governmental discourses than in funding research calls, and occasionally include explicit references to vulnerability, depicting vulnerable populations as those in need of (governmental) support to achieve resilience in relation to food insecurity and cost-of-living (the main contexts of use of 'resilience' in this study). Focus group participants also commented on the importance of social support to achieve resilience, although we observed a tension between understandings of resilience as individual skill and autonomy, and resilience as the capacity to ask for help and achieved through the community. It is unclear, however, who is identified as help provider (e.g., family, third sector, or the government), revealing an important tension in the understanding of the concept.

Other tensional meanings identified include the association of resilience with the restoration of a previous (better) state, particularly observed in focus groups, and resilience as capacity to adapt, the latter associated with the economy and environment in governmental and funders' discourses, and with the individual in focus groups. These two meanings echo the distinction noted in Holling's [37] between system stability or permanence, and system resilience as flexibility and adaptability. This second understanding, in line with the recent reconceptualizations advanced by Resilience Engineering, is associated with sustainability in governmental and funders' discourses, which allude to sustainable development as a means to achieve climate change resilience by developing technology to lower carbon emissions and building new supply chains. Resilience is characterised as making the necessary adjustments to keep the current economic and social activities in the new environmental circumstances; mentions to the (potential) need of socioeconomic changes which may involve different relations with nature have not been identified.

In a similar manner, while the governmental discourses identify some social groups as being particularly vulnerable to the food and cost-of-living crisis, resilience is articulated as the provision of financial help in the form of governmental support schemes. Thus, resilience is

constructed as offering support to disadvantaged communities through the system of benefits to palliate pre-existing inequalities. While this financial support is in much need, these practices present resilience as acquiring the capacity to endure economic (or environmental) struggles, which contrasts with adopting preventive measures to tackle the inequalities (or practices) that lead to them.

As shown in the focus groups, the public may see resilience as the responsibility of individuals, or at least for individuals to seek appropriate support; the question is: who should be offering this support? In the context of cost-of-living, the centrally awarded, locally administered Household Support Fund has been extended (at the time of writing) for six months, while the primary response to household food insecurity remains based in the third sector, in food aid (which may or may not receive support from local government). There needs to be policy coherence in terms of the source of funds/resources to enable resilience.

While focus groups participants commented on the inadequacy of applying resilience in the health context, observing that health conditions are beyond the sufferers' control, criticism towards the use of resilience in relation to communities facing economic hardships, or in relation to the environmental crisis, were not identified. However, participants recognised resilience as a vague concept subject to multiple interpretations, and potential misinterpretations, in the absence of definitions or examples. These observations converge with our analysis findings: the different communities represented in the data link to different orders of indexicality [4], and the influences of academic and policy discourses on the term 'resilience' are not necessarily accessible to the public. Contrary to other political keywords, which tend to index their discourse community or political orientation, 'resilience' seems to involve 'layered simultaneity' [4] of discourse history indexicalities. While policy making has adopted some of the scientific meanings, the public still connects to older layers of meaning represented in the BNC.

## 6.2 Limitations and future research directions

Findings of this study are restricted to the UK context; populations with different cultural, sociopolitical and economic traditions may show different understandings and applications of the concept. Importantly, the limited number of focus group participants makes it difficult, if not impossible, to extrapolate findings to the general population. Lack of demographic information does not make it possible to examine whether understandings of resilience vary across different social groups—e.g. some focus group participants observed that different age groups show different orientations towards the concept, with elder populations conceptualising resilience in relation to material conditions while young populations understand it in relation to psychological wellbeing and the individual ability to seek support. Follow-up studies on lay understandings of the concept should complement focus groups with other public involvement methods which allow for more efficient data gathering and the possibility to scale-up findings, such as questionnaires including both closed and open questions (see, for example, [62, 63] for mixed-methods studies combing corpus linguistics with representative surveys applied to public health guidance reception).

While this study specifically focused on examining uses of resilience in governmental discourses and funding calls to identify possible correspondences between these two domains, restricting the data to these registers has inevitably limited the uses of 'resilience' to those that appear in these texts. Examining uses of 'resilience' in health communication discourses or news media would provide a more encompassing overview of the different conceptualisations and areas of application of the concept, illustrating how specific disciplines adopt it (e.g. health sciences vis-à-vis politics) and how it is communicated to the public.

### 6.3 Recommendations

Despite the restrictions in the registers considered, this study has evidenced that 'resilience' allows for a cacophony of meanings which can lead to confusion when applied across domains. We suggest the need of specificity in public policy and official communications around resilience if the concept is to be used at all, including communications for research communities, since resilience may be understood differently across disciplines. Our analysis also rises a cautionary note about the potential inferences of moral judgements that resilience may allow for if adopted to describe individuals' character or behaviours, which may contribute to discourage collaborative attitudes across the public.

Finally, this study raises the question about the adoption of political language into research funding calls. Evidence generation through research is supposed to inform policy, however how policy—shaped by political actors—informs communication of research priorities is an area worth critically studying. Shaping the research agenda should ideally be evidence-driven itself and studying how political language potentially directs it may benefit the greater societal good and optimise the cost benefit of tax-payer funded research.

## Supporting information

**S1 File. Appendix 1 –Table 7.** 50 first keywords ('terms') ranked by keyness score and classified in themes.
(DOCX)

**S2 File. Appendix 2 –Table 8.** Collocates of "resilience"; S2 File. Appendix 2 –Table 9. Collocates of "resilient".
(DOCX)

## Author Contributions

**Conceptualization:** Sara Vilar-Lluch, Donna Clutterbuck, Michael Kranert, Dianna Smith, Sarah Nield, Nisreen A. Alwan.

**Data curation:** Sara Vilar-Lluch, Donna Clutterbuck.

**Formal analysis:** Sara Vilar-Lluch, Donna Clutterbuck, Michael Kranert.

**Funding acquisition:** Michael Kranert, Dianna Smith, Sarah Nield, Nisreen A. Alwan.

**Investigation:** Sara Vilar-Lluch, Donna Clutterbuck, Michael Kranert, Dianna Smith, Sarah Nield, Nisreen A. Alwan.

**Methodology:** Sara Vilar-Lluch, Donna Clutterbuck, Michael Kranert, Dianna Smith, Nisreen A. Alwan.

**Project administration:** Nisreen A. Alwan.

**Supervision:** Nisreen A. Alwan.

**Writing – original draft:** Sara Vilar-Lluch.

**Writing – review & editing:** Sara Vilar-Lluch, Donna Clutterbuck, Michael Kranert, Dianna Smith, Sarah Nield, Nisreen A. Alwan.

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
