## [Decision Letter · Decision Letter 0]

6 Aug 2024

PONE-D-24-22466Being “resilient” and achieving “resilience”: from governmental discourses to the national research agenda in the contexts of food insecurity and cost of livingPLOS ONE

Dear Dr. Alwan,

Thank you for submitting your manuscript to PLOS ONE. After careful consideration, we feel that it has merit but does not fully meet PLOS ONE’s publication criteria as it currently stands. Therefore, we invite you to submit a revised version of the manuscript that addresses the points raised during the review process.

We look forward to receiving your revised manuscript.

Kind regards,

Andrea Tomassi, Ph.D.

Academic Editor

PLOS ONE

Journal Requirements:

2. In this instance it seems there may be acceptable restrictions in place that prevent the public sharing of your minimal data. However, in line with our goal of ensuring long-term data availability to all interested researchers, PLOS’ Data Policy states that authors cannot be the sole named individuals responsible for ensuring data access (http://journals.plos.org/plosone/s/data-availability#loc-acceptable-data-sharing-methods).

Additional Editor Comments:

Dear Authors,

Your manuscript "Being ‘resilient’ and achieving ‘resilience’: from governmental discourses to the national research agenda in the contexts of food insecurity and cost of living" presents an ambitious and timely analysis of the evolution of the term 'resilience' within UK policy discourse. The topic is undoubtedly significant, and your methodological approach using corpus-based discursive analysis alongside focus group discussions adds a rich dimension to the exploration. However, as indicated by the reviewers, there are several areas where improvements are necessary to enhance the clarity, depth, and impact of your work.

Conceptual and Theoretical Expansion:

Reviewer 1 suggests that your historical overview of 'resilience' lacks a discussion on organizational resilience and resilience engineering. These areas, particularly as developed by scholars like Erik Hollnagel, could significantly deepen the theoretical grounding of your paper. Incorporating these perspectives could provide a more comprehensive view of how resilience functions not just at individual or community levels but also within complex socio-technical systems.

Methodological Details:

Both reviewers have pointed out that there is a need for more detailed methodological transparency. Reviewer 2, in particular, mentions the misalignment between the title, introduction, and the study's objectives. Clarifying how each section of your manuscript contributes to your stated aims will help strengthen your argument's logical flow. Moreover, expanding on your focus group methodology—such as the criteria for participant selection and the rationale behind the chosen questions—will enhance the validity of your findings.

Addressing Selection Bias:

The concern about selection bias, as raised by Reviewer 1, is crucial. Your focus on specific elements of resilience within governmental discourse might benefit from a broader lens. Consider how you might include additional data sources or expand your analytical framework to mitigate this bias. This could involve examining a wider range of documents or incorporating more diverse viewpoints from different sectors or geographical areas within the UK.

Enhancing the Relevance to the British Context:

There is a noted need for justifying the geographical focus on the UK more robustly. Discuss why the UK presents a unique or particularly informative case for studying resilience in the context of food insecurity and cost of living. This could involve more explicitly linking your findings to current debates or policy initiatives within the UK.

Refining Language and Definitions:

Ensure that the term 'resilience' is consistently defined and used throughout the paper. As resilience is a multi-dimensional concept, a clear and operationalized definition at the beginning of your paper will help maintain focus and prevent the ambiguity noted by Reviewer 1.

Your study has the potential to contribute significantly to the discourse on resilience in policy-making. By addressing these comments and incorporating the detailed feedback provided by the reviewers, your paper can achieve greater analytical depth and theoretical impact. I look forward to your revised manuscript.

Best regards

Reviewers' comments:

Reviewer's Responses to Questions

**Comments to the Author**

1. Is the manuscript technically sound, and do the data support the conclusions?

Reviewer #1: Partly

Reviewer #2: Partly

2. Has the statistical analysis been performed appropriately and rigorously? 

Reviewer #1: No

Reviewer #2: N/A

3. Have the authors made all data underlying the findings in their manuscript fully available?

Reviewer #1: No

Reviewer #2: Yes

4. Is the manuscript presented in an intelligible fashion and written in standard English?

Reviewer #1: Yes

Reviewer #2: Yes

5. Review Comments to the Author

Reviewer #1: In "Being ‘resilient’ and achieving ‘resilience’: from governmental discourses to the national research agenda in the contexts of food insecurity and cost of living," the authors aim to explore and define the concept of ‘resilience’ within the contexts of food insecurity and the cost of living, specifically examining its representation in UK governmental discourses and the influence of these policy representations on research funding calls by UK national bodies. The objective is to demonstrate how the term ‘resilience’ has evolved from an individual psychological trait to a socio-economic and environmental characteristic, analyzing how these representations may shape research orientations and public communication. The investigation employs a corpus-based discursive analysis of government documents and research funds, complemented by focus group discussions, to compare official representations of ‘resilience’ with public understandings.

The article is impeccably presented, well-written, and elegantly structured. The authors address a pressing issue: the unspecified overuse of the term resilience, which in the hands of unaware individuals becomes a blunt conceptual tool, reduced to an umbrella term. While I personally understand the intentions behind this work and partially agree with its reflections, I believe the authors have fallen into a selection bias that may compromise the validity of their findings. The article focuses on a concept known for its multidimensional nature and varying interpretations depending on the context. The lack of a clear and operationalized definition of "resilience" uniformly applied throughout the analysis can lead to ambiguous or less rigorous conclusions. I believe this is the case here.

The authors, in their brief and unfortunately not exhaustive historical overview, cite seminal works by McGreavy, Frydenberg, Woods, and Holling, tracing the evolution of the term resilience from its original Latin meaning through its usage in cognitive and ecological sciences. However, they omit an important semantic facet recognized by a growing academic community, which could have been uncovered and analyzed by further investigating Woods' work: organizational resilience or resilience in the sense of Resilience Engineering. In the context of this study, this perspective would have offered an integrated and fundamental viewpoint on the subject. This vision, developed by experts like Erik Hollnagel, focuses on the ability of socio-technical systems to maintain safe and continuous operations in uncertain and changing contexts. Hollnagel defines resilience as the capacity of a system, whether technical, human, or organizational, to adapt, absorb, and rapidly recover from disruptions or changes. This vision emphasizes four main capabilities: responding, monitoring, anticipating, and learning. These capabilities apply not only to individuals but also to collectives and organizations, demonstrating how resilience can be a property of entire socio-technical systems.

The evolution of the concept of resilience as described by Resilience Engineering is a direct response to the inadequacy of traditional safety and security concepts, which no longer meet the needs of modern complex socio-technical systems. These systems, which include critical infrastructures, complex organizations, and global networks, require a more dynamic and adaptive approach to manage uncertainty and disruptions. The concept of complexity and complex adaptive systems (CAS) offers a crucial perspective on resilience, highlighting how it emerges from both internal interactions and external relationships. This approach not only challenges the researchers' partial and potentially distorted interpretations but also demonstrates that resilience is an intrinsically dynamic and multifactorial quality. The coexistence of internal and external forces makes the described framework more coherent and diminishes the strength of their critiques, underscoring the importance of considering resilience as an emergent and adaptive property in modern socio-technical systems.

To claim that resilience should not be attributed to collective agents is reductive. Communities, organizations, and entire nations must be resilient to face global challenges. Individual psychological resilience is just one component; collective resilience is crucial to ensure that complex systems can adapt and thrive. Critical analyses that ignore this collective dimension risk promoting a limited and unrepresentative view of reality. Consequently, the presence of resilience in the political and research agendas is not only justified but necessary. Global challenges such as climate change, economic crises, and pandemics require a systemic approach. In this context, resilience becomes an essential framework for developing policies and practices that enable social, economic, and technical systems to confront and overcome disruptions.

The authors chose to use governmental discourses and funding calls as their primary sources. Limiting the analysis to these documents may have reinforced the selection bias, inadvertently consolidating a frame where the discourse of safety and security, and thus the collective behavior of social, technological, and research systems (including family management as an organization), simply does not exist. This greatly limits the various facets and contexts in which the concept of "resilience" is applied. Additionally, the number of focus groups conducted and the demographic composition of participants are not specified. If the groups are not representative of the general population or specific subgroups, the results may not be generalizable. This could be an interesting reflection point for the authors: research might guide policymakers through an inappropriate lexicon, leading to policies implemented without adequate communication. Perhaps this could be the true theme of the article.

While the use of corpus-based analysis is a solid approach for studying linguistic usage in large datasets, the validity of the results heavily depends on the tools and analysis procedures employed. Without details on the criteria for selecting keywords, statistical significance metrics, and normalization procedures, there might be concerns about the accuracy of the analysis. Relying uncritically on standard linguistic analysis software solutions only heightens my doubts. The choice of collocation analysis seems appropriate, less so the relative frequency analysis. Sometimes, as you surely know, the presence of hapax legomena can be more explanatory than frequent words for interpreting text semantically. Moreover, an analysis of the lexical richness of both the documents and the linguistic capacity of the target referent might be more appropriate in this context.

Ultimately, it seems that the authors have (re-)discovered an additional facet of the construct "resilience" according to resilience engineering. Resilience engineering provides a framework that recognizes the complexity and adaptive nature of modern systems. Instead of viewing resilience as a simple individual recovery capacity, resilience engineering considers it an emergent property of complex systems, where dynamic interactions and interdependencies between internal and external components play a crucial role. Resilience engineering aligns with constructivist approaches, which see knowledge and reality as socially constructed through continuous interaction between human and non-human actors. This approach acknowledges that resilience is not a static characteristic but a continuous process of adaptation and transformation.

Actor Network Theory (ANT) further contributes to this vision, emphasizing that agency is distributed among networks of both human and non-human actors. According to ANT, resilience emerges from the relationships among these actors, including individuals, technologies, infrastructures, and institutions. This perspective broadens the concept of resilience beyond the individual level, encompassing the entire network of interconnected actors.

When reinterpreted through the lens of resilience engineering and ANT, research themes and the political agenda become more coherent and relevant. Here's how:

- Critical Infrastructure Systems: Research on the resilience of critical infrastructures, such as energy, water, and transportation, becomes fundamental. Instead of focusing solely on incident prevention, resilience engineering promotes adaptability and recovery capabilities through dynamic design and management.

- Environmental and Climate Policies: Environmental policies aimed at combating climate change gain a new dimension. Resilience engineering encourages the adoption of sustainable and adaptive practices that allow communities and ecosystems to effectively deal with climatic disruptions.

- Socio-Economic Development: Economic and social resilience is viewed through the prism of interactions among various actors. Policies promoting socio-economic resilience must consider the interdependencies between communities, institutions, and technologies, fostering networks of support and collective innovation.

- Public Health: In public health, resilience engineering suggests approaches that strengthen health systems through continuous learning and adaptation, recognizing the crucial role of networks of involved actors, from health institutions to patients, from medical technologies to health policies.

For these reasons, if the authors wish to publish their work, they are strongly encouraged to repeat their analyses by integrating the interpretative facet they missed. Otherwise, the result, being significantly distorted, cannot be published.

Reviewer #2: The article is a good contribution to theoretical research about the concepts of resilience in the contexts of food insecurity and cost of living in the UK. Please address the following comments and, in general, try to provide better details about your methodology and research decisions, it will increase the article's value.

1. There is a shift in the objective between the title and the introduction and research questions. The title and the query used in the methodology focus on the topic of food insecurity and cost of living, whereas the introduction does not address this topic. Please clarify this point by introducing the topic of food insecurity and cost of living in the introduction to align it with the title and methodology.

2. The study focuses on a British context. It is important to introduce and justify the relevance of this geographical focus. Explain why the British scenario is significant for the study and eventually address limitations for this focus.

3. The focus group target appears to be quite generic. Are the participants experts in food insecurity and/or cost of living? How can their expertise in the topic of resilience be assessed? Provide clear criteria or justification for the selection of focus group participants to ensure their relevance to the study.

4. The questions addressed to the focus groups are very generic and not specifically focused on food, which makes the analysis not comparable to the Corpus-based Discourse Analysis and Critical Policy Discourse Analysis. Justify the choice of questions and explain how they align with the objectives of the study.

5. The term “example prompt” used in Table 2 is unclear and there are no references to this in the article. Clarify what is meant by “example prompt”.

6. The topic of COVID-19 emerged in the focus group discussions, which is certainly linked with resilience and food insecurity. There have been multiple research studies addressing how the concept of resilience has adapted to the pandemic and the topic of food insecurity and food supply chains. Add a section in the background discussing the impact of COVID-19 on resilience and food insecurity, citing relevant studies to provide a comprehensive context.

7. The authors have only analyzed the absolute frequency of terms. It might be interesting to also analyze term frequency-inverse document frequency (tf-idf) to identify if some specific documents or topics are more relevant. Highlighting the importance of terms in specific documents relative to their occurrence in the entire dataset can provide deeper insights into the data.

8. The study lacks a discussion on future research directions and limitations. Include a section that addresses the limitations of the current study and suggests areas for future research to provide a pathway for further investigation and to acknowledge the boundaries of the current research.

6. PLOS authors have the option to publish the peer review history of their article (what does this mean?). If published, this will include your full peer review and any attached files.

Reviewer #1: No

Reviewer #2: No

---

## [Author Response · Author response to Decision Letter 0]

16 Oct 2024

Please see attached response to editor/reviewers document.

---

## [Decision Letter · Decision Letter 1]

9 Dec 2024

Being “resilient” and achieving “resilience”: from governmental discourses to the national research agenda in the contexts of food insecurity and cost of living

PONE-D-24-22466R1

Dear Dr. Alwan,

We’re pleased to inform you that your manuscript has been judged scientifically suitable for publication and will be formally accepted for publication once it meets all outstanding technical requirements.

Kind regards,

Laura Kelly, PhD

Division Editor

PLOS ONE

Additional Editor Comments (optional):

Reviewers' comments:

Reviewer's Responses to Questions

**Comments to the Author**

1. If the authors have adequately addressed your comments raised in a previous round of review and you feel that this manuscript is now acceptable for publication, you may indicate that here to bypass the “Comments to the Author” section, enter your conflict of interest statement in the “Confidential to Editor” section, and submit your "Accept" recommendation.

Reviewer #2: All comments have been addressed

2. Is the manuscript technically sound, and do the data support the conclusions?

Reviewer #2: Partly

3. Has the statistical analysis been performed appropriately and rigorously? 

Reviewer #2: N/A

4. Have the authors made all data underlying the findings in their manuscript fully available?

Reviewer #2: Yes

5. Is the manuscript presented in an intelligible fashion and written in standard English?

Reviewer #2: (No Response)

6. Review Comments to the Author

Reviewer #2: The authors have addressed the reviewer's comments properly. The article has improved and can be accepted for publication

7. PLOS authors have the option to publish the peer review history of their article (what does this mean?). If published, this will include your full peer review and any attached files.

Reviewer #2: No

---

## [Editor Report · Acceptance letter]

20 Dec 2024

PONE-D-24-22466R1 

PLOS ONE

Dear Dr. Alwan, 

I'm pleased to inform you that your manuscript has been deemed suitable for publication in PLOS ONE. Congratulations! Your manuscript is now being handed over to our production team.

Kind regards, 

on behalf of

Dr. Laura Hannah Kelly 

Staff Editor

PLOS ONE